# Exploring the mitochondrial genomes and phylogenetic relationships of trans-Andean Bryconidae species (Actinopterygii: Ostariophysi: Characiformes)

**Edna Judith Márquez**[1]*, **Daniel Alfredo Gómez-Chavarría**[2], **Juan Fernando Alzate**[2]

**1** Laboratorio de Biología Molecular y Celular, Facultad de Ciencias, Universidad Nacional de Colombia, Sede Medellín, Medellín, Antioquia, Colombia, **2** Centro Nacional de Secuenciación Genómica-CNSG, Facultad de Medicina, Universidad de Antioquia, Medellín, Antioquia, Colombia

☯ These authors contributed equally to this work.
* ejmarque@unal.edu.co

**Data Availability Statement:** All mitogenomes sequences files are available from the GenBank database: Brycon meeki: (OR168942), Brycon moorei: (OR168957), Brycon oligolepis:

## Abstract

Comparative mitogenomics and its evolutionary relationships within Bryconidae remains largely unexplored. To bridge this gap, this study assembled 15 mitogenomes from 11 Bryconidae species, including five newly sequenced. *Salminus* mitogenomes, exceeding 17,700 bp, exhibited the largest size, contrasting with a median size of 16,848 bp in the remaining species (*Brycon* and *Chilobrycon*). These mitogenomes encode 37 typical mitochondrial genes, including 13 protein-coding, 2 ribosomal RNA, and 22 transfer RNA genes, and exhibit the conserved gene arrangement found in most fish species. Phylogenetic relationships, based on the maximum-likelihood method, revealed that the trans-Andean species (found in northwestern South America) clustered into two main sister clades. One clade comprised the trans-Andean species from the Pacific slope, *Brycon chagrensis* and *Chilobrycon deuterodon*. The other clade grouped the trans-Andean species from the Magdalena-Cauca Basin *Brycon moorei* and *Salminus affinis*, with their respective cis-Andean congeners (found in eastern South America), with *Brycon rubricauda* as its sister clade. Since the current members of *Brycon* are split in three separated lineages, the systematic classification of Bryconidae requires further examination. This study provides novel insights into mitogenome characteristics and evolutionary pathways within Bryconidae, standing as crucial information for prospective phylogenetic and taxonomic studies, molecular ecology, and provides a valuable resource for environmental DNA applications.

## Introduction

Mitochondrial DNA is considered a valuable molecular tool for teleostean fishes genetic diversity studies as well as for addressing interspecific and intraspecific evolutionary relationships due to its rapid evolutionary rate, short coalescence times, high gene diversity, low molecular weight, haploidic condition, and uniparental inheritance [1–5]. The fish mitogenome shows a

(OR168943), Brycon rubricauda: (OR168944), Salminus affinis: (OR168945), Brycon amazonicus: (OR168948), Brycon amazonicus: (OR168947), Brycon chagrensis: (OR168949), Brycon chagrensis: (OR168950), Brycon falcatus: (OR168952), Brycon falcatus: (OR168951), Brycon orbignyanus: (OR168953), Chilobrycon deuterodon: (OR168954), Salminus brasiliensis: (OR168956), Salminus brasiliensis: (OR168955).

**Funding:** This study was supported by a grant framed under the Project "Variabilidad genética de un banco de peces de los sectores medio y bajo del río Cauca" (CT-2019-000661, Empresas Públicas de Medellín and Universidad Nacional de Colombia, Sede Medellín). Funders do not play any role in the study design, data collection and analysis, decision to publish, or preparation of the manuscript.

**Competing interests:** The authors have declared that no competing interests exist.

highly conserved gene array, although some exceptions have been detected [6]. Since the differential evolutionary rate of its genes may reveal different evolutionary histories, compared with the partial sequences of a few mitochondrial genes [5], the fishes mitogenome has been used to solve inter- and intraspecific relationships [7–9], historical biogeography [10], evolutionary origin [11, 12], and comparative mitogenomics [6, 13], among other aplications.

Bryconidae comprises five main clades arranged in four genera *Brycon*, *Chilobrycon*, *Henochilus*, and *Salminus* [14]. *Brycon* is non-monophyletic [14] and encompasses 44 valid species distributed from southern Mexico to northern Argentina [15], being also a key economical resource in Central and South America by supporting subsistence and commercial fisheries, sport fishing, and aquaculture [16]. However, unresolved taxonomic problems and highly divergent mitochondrial lineages of *Brycon* stimulate the search for more informative genes [17]. *Chilobrycon* and *Henochilus* are two monotypic genera restricted to the Pacific slope of northern Peru and Ecuador, and eastern Brazil, respectively. *Salminus* comprises six species with distribution across the main basins of South America: Amazon, Orinoco, Paraná-Paraguay, São Francisco, and Magdalena River basins [18].

Additionally, the controversial phylogenetic relationships within the family and among families of the order Characiformes [19, 20] and taxonomic sampling incompleteness in phylogenetic analysis stimulate the necessity to obtain new information that provides insights into its evolutionary history. Comparative mitogenomics and its evolutionary relationships among Bryconidae remains unexplored so far. Consequently, in line with the idea that a wider sampling of taxa and individuals is required to improve the relationships understanding within Bryconidae, this study provides the complete mitogenome of five species from north-western South America, *Brycon meeki* Eigenmann & Hildebrand, 1918, *Brycon moorei* Steindachner, 1878, *Brycon oligolepis* Regan, 1913, *Brycon rubricauda* Steindachner, 1879 and *Salminus affinis* Steindachner, 1880. The four complete mitogenomes already available so far include *Brycon orbignyanus* (Valenciennes 1850) [21], *Brycon henni* Eigenmann, 1913 [22], *Brycon nattereri* Günther 1864 [23], and *Salminus brasiliensis* (Cuvier 1816) [24]. This study further assembled, by data mining, 10 mitogenomes from other species including *Brycon amazonicus* (Agassiz, 1829), *Brycon chagrensis* (Kner 1863), *Brycon falcatus* Müller & Troschel, 1844, *Brycon orbignyanus*, *Chilobrycon deuterodon* Géry & de Rham 1981, and *S. brasiliensis*.

## Material and methods

This study assembled a total of 15 mitogenomes corresponding to 11 species of Bryconidae. For obtaining and sequencing the mitochondrial genome of five species from north-western South America, this study analyzed muscle or caudal fin samples preserved in 95% ethanol from *Brycon moorei*, *B. rubricauda* and *Salminus affinis*, collected in the middle and lower Cauca River by the Universidad de Córdoba and the Universidad de Antioquia. This study also included caudal fin samples preserved in 70% ethanol of *B. meeki* and *B. oligolepis*, collected in the Anchicayá River, Pacific slope.

Isolation of total genomic DNA from tissues was performed with the QIAamp DNA Mini Kit (Qiagen), following the manufacturer's recommendations for muscle tissue. DNA integrity was evaluated by agarose gel electrophoresis, and its concentration was quantified by light absorption at 260nm using the NanoDrop™ 2000-Thermo Scientific™ and the Picogreen fluorescent method. The Next Generation Sequencing (NGS) for *S. affinis* and the north-western *Brycon* species was performed on an Illumina MiSeq instrument reading 300 paired end reads. Whole genome shotgun libraries were prepared with the Illumina Truseq Nano DNA kit. Raw reads were filtered using the CUTADAPT software v2.10 [25], eliminating remaining Truseq adapter sequences, read ends below Q30 quality threshold, and reads with ambiguous bases.

**Table 1. List of mitogenomes of Bryconidae included in this study.**

| Species | Sample Name | SRA Run | GenBank Accession | Country | Source |
|---|---|---|---|---|---|
| *Brycon meeki* | Bme008 | NA | OR168942 | Colombia | This study |
| *Brycon moorei* | C19274 | NA | OR168957 | Colombia | This study |
| *Brycon oligolepis* | Bol019 | NA | OR168943 | Colombia | This study |
| *Brycon rubricauda* | A183 | NA | OR168944 | Colombia | This study |
| *Salminus affinis* | C8284 | NA | OR168945 | Colombia | This study |
| *Brycon amazonicus* | LBP-14082 | SRR10832392 | OR168948 | Brazil | SRA |
| *Brycon amazonicus* | 58483 | SRR10079810 | OR168947 | NA | SRA |
| *Brycon chagrensis* | 28B05 | SRR11587716 | OR168949 | Panama | SRA |
| *Brycon chagrensis* | 29B07 | SRR11587715 | OR168950 | Panama | SRA |
| *Brycon falcatus* | 53445 | SRR10079809 | OR168952 | NA | SRA |
| *Brycon falcatus* | 53437 | SRR10079808 | OR168951 | NA | SRA |
| *Brycon orbignyanus* | Voucher120457 | SRR6243207 | OR168953 | Brazil | SRA |
| *Chilobrycon deuterodon* | 45001 | SRR10079817 | OR168954 | NA | SRA |
| *Salminus brasiliensis* | SZAIPI037396-87 | SRR17407720 | OR168956 | Brazil | SRA |
| *Salminus brasiliensis* | 21905 | SRR10079814 | OR168955 | NA | SRA |
| *Brycon henni* | Bhen-UNAL-001 | NA | NC_026873.1 | Colombia | GenBank |
| *Brycon nattereri* | LAGEEVO_3928 | NA | NC_051927.1 | NA | GenBank |
| *Brycon orbignyanus* | NA | NA | NC_024938.1 | NA | GenBank |
| *Salminus brasiliensis* | NA | NA | NC_024941.1 | Brazil | GenBank |

The samples from northwestern South America are shaded in gray. A total of 15 mitogenomes (trans-Andean: 8; cis-Andean: 7) were assembled in this study from newly sequencing data (5) and SRA data (10).

Genomic assembly was performed with SPADES assembler v3.14.1 [26], using default parameters. The scaffold containing the mitochondrial genomes was detected using BLASTN [27] and customed database of fish mitochondrial genomes.

The remaining 10 mitochondrial genomes were generated by downloading NGS genomic or transcriptomic data from the Sequence Read Archive (SRA) database, followed by read cleaning and subsequent assembly (see Table 1). The sample listed under the SRA accession SRR10079810 was originally labeled as *Brycon falcatus*, while the library name was labeled as brycon_amazonicus58483 (https://www.ncbi.nlm.nih.gov/sra/?term=SRR10079810). However, the phylogenomic analysis confirmed its correct taxonomic position as *B. amazonicus*, as listed in Table 1.

The mitochondrial genomes were annotated using the MITO-ANNOTATOR TOOL of the MITOFISH webserver v3.86 (10.1093/molbev/msy074). The synteny of the mitogenomes was assessed using the MAUVE genome aligning and visualization tool [28]. The other new reference mitogenomes were generated by downloading and assembling NGS raw read data, available at the SRA database, as described in Table 1. Read quality filtering, genomic assembly, and mitochondrial genome detection and annotation were performed as described above for the DNA-seq experiments. As for the RNA-seq data of *B. falcatus*, *de novo* transcriptome assembly was performed with the Trinity package [29], applying default parameters. NGS reads were mapped against the respective mitogenome scaffold using BOWTIE2, and the average sequencing depth was calculated to assess the sequencing coverage obtained for the newly generated mitochondrial genomes using SAMTOOLS software. The strand asymmetry was determined using the formula AT skew = (A-T)/(A+T) [30].

For comparative and evolutionary analysis purposes, this study included four mitogenomes previously published for the following species: *Salminus brasiliensis* [24], *Brycon orbignyanus*

[21], *B. nattereri* [23], and *B. henni* [22]. As outgroups, this study included the mitogenome of the related species *Triportheus magdalenae* (Tmagd001; GenBank accession OR168946), which was sequenced and assembled as described above for the other species, *Prochilodus vimboides* (NC037712), and *Chalceus macrolepidotus* (NC004700).

A Principal Component Analysis (PCA) was conducted by examining variant features within the genomic annotations of Bryconidae mitogenomes, including gene lengths (CDS, rRNA, tRNA), and D-loop region lengths. A comprehensive table encompassing all genome features exhibiting variations in base pair lengths was imported into the R statistical package, [31] and underwent processing using the scale and prcomp functions. The resulting geometric point graph was generated utilizing ggplot2.

The phylogenetic relationships among *Brycon* species from northwestern South America and other Bryconidae species were inferred using the maximum likelihood method with the IQTREE2 program. This study constructed a super matrix consisting of 15 mitochondrial genes, including 13 CDSs and two mitochondrial rRNAs. Each gene individually extracted was aligned with its respective homologous loci using MAFFT. Subsequently, the 15 individual alignments were concatenated with the program *catsequences* (https://github.com/ChrisCreevey/catsequences). IQTREE2 running parameters included the partitions option, treating each individual gene as a partition, the search for the best substitution model for each partition, and 5000 ultra-fast bootstrap (UFB) pseudo replicates. Additionally, this study calculated two concordance factors [32, 33]: gene concordance factor (GCF) and site concordance factor (SCF). The tree visualization and graphical editing were performed in the FIGTREE program v1.4.4 (http://tree.bio.ed.ac.uk/software/figtree/).

## Results

The Bryconidae mitogenomes (Figs 1 and 2) showed the same mitochondrial genome structure and gene arrangement, although *trn*F and the D-loop sequences were not recovered for one of the *B. chagrensis* mitogenome (29B07). The protein coding gene *nad6* and eight tRNAs (*trn*A, *trn*C, *trn*E, *trn*N, *trn*P, *trn*Q, *trn*S2, *trn*Y) are coded by the N-strand, while the remaining genes, including the D-loop region, are coded by the J-strand (*nad1*, *nad2*, *cox1*, *cox2*, *atp6*, *atp8*, *cox3*, *nad3*, *nad4l*, *nad4*, *nad5*, *cytb*, 14 tRNAs and 2 rRNAs).

The median sequencing depth for the 15 mitogenome scaffolds was 134X, with values ranging from 4785X to 23X, for *Brycon chagrensis* 29B07 and *B. meeki*, respectively (Table 2). The higher coverage values, both over 4000X, were obtained for the *B. chagrensis* scaffolds that came from RNA-seq data. The GC content ranged from 41.96% to 44.88% (Median: 43.68%) and the overall AT skews ranged from 0.015 to 0.071 (Median: 0.057).

In terms of mitochondrial genome sizes (Table 2), the largest were the *Salminus* mitogenomes (over 17,700 bp; Median: 17,799 bp) compared with the other species (*Brycon* and *Chilobrycon*, median mitogenome size: 16,848 bp). The Bryconidae mitogenomes from northwestern South America (Mean: 16894.29; Median: 16,884) were larger than those from southeastern South America (Mean: 16823.57; Median: 16,837; p = 0.011). The *B. meeki* mitogenome was similar in length to *B. chagrensis*, followed by *B. henni*, *B. oligolepis*, *Ch. deuterodon*, *B. rubricauda* and *B. moorei* (Table 3). Regarding the gene lengths variation (Table 3), *Ch. deuterodon* showed notable differences with the other Bryconidae mitogenomes in D-loop, mt-rnr2, and nd6, whereas *B. moorei* showed variation patterns more similar to *B. orbignyanus* from southeastern South America. In general, the gene length was more similar in mitogenomes of southeastern South America species. The PCA of gene lengths variation showed that the first principal component, which explains 34.3% of the total variation, separates *Salminus* from *Brycon* and *Chilobyrcon* mitogenomes (Fig 3). Additionally, the second principal

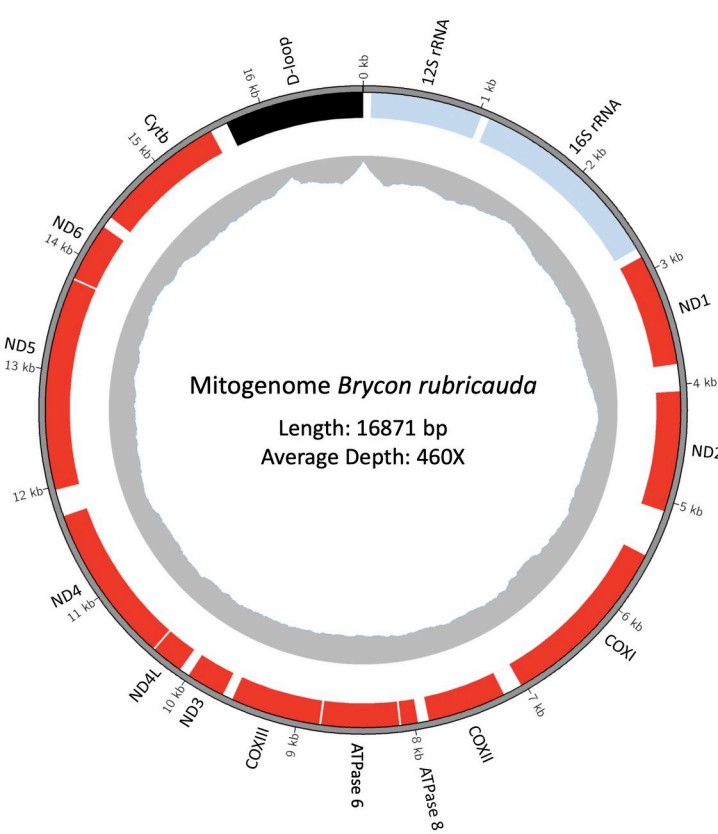

**Fig 1. Mitochondrial genome of *Brycon rubricauda* spanning 16,871 base pairs (bp).** The outermost gray ring represents the DNA molecule and serves as a kilobases (kb) size scale. Protein-coding sequences (CDS) are highlighted in red boxes, rRNA genes are denoted in light blue, and the D-loop region is marked in black. The innermost gray plot represents the sequencing depth, indicating a median coverage of 460X.

component, which explains 25% of the total variation, separates trans-Andean from cis-Andean species, except *B. moorei* and *S. affinis* that were clustered with cis-Andean species.

The greatest differences in length among non-protein-coding genes or regions were observed in the D-loop region (*Ch. deuterodon*: 1113; *Brycon*: 1,099–1285; *Salminus*: 2128–2574) even within the same species (*S. brasiliensis*, *B. orbignyanus*, and *B. amazonicus*,

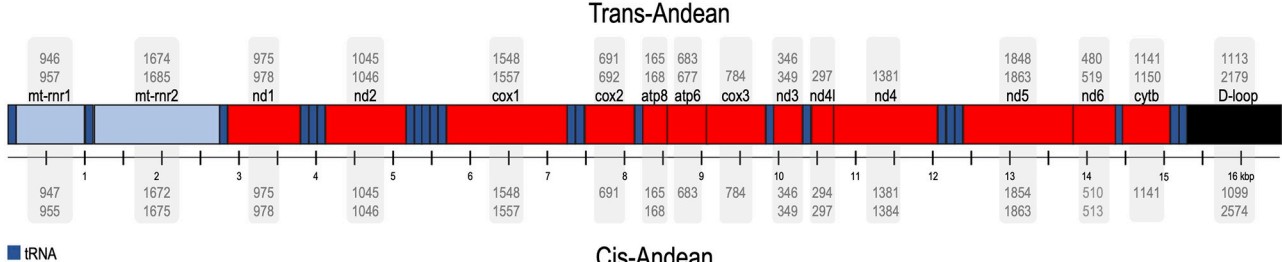

**Fig 2. Synteny and structural variation analysis of mitochondrial genomes in the Bryconidae family.** The mitochondrial genome structure within the Bryconidae family reveals conserved synteny across all members, accompanied by variations in gene and D-loop region sizes. Coding genes are highlighted in red, ribosomal RNA genes in light blue, transfer RNA genes in dark blue, and the D-loop region in black. Genome feature lengths are presented as ranges from minimum to maximum values. Single values denote genes maintaining consistent lengths across all analyzed species. Feature lengths for Trans-Andean species are located at the top of the Figure while their cis-Andean counterparts are at the bottom.

**Table 2. Summarized mitogenomic and assemble characteristics of the 13 Bryconidae species.**

| Species | Sample Code | Average coverage | GC % | AT skew | Length (bp) |
|---|---|---|---|---|---|
| *Brycon amazonicus* | LBP-14082 | 35 | 42.51% | 0.062 | 16811 |
| *Brycon amazonicus* | 58483 | 36 | 42.59% | 0.064 | 16754 |
| *Brycon chagrensis* | 28B05 | 4077 | 44.06% | 0.055 | 16940 |
| *Brycon chagrensis* | 29B07 | 4785 | 44.88% | -0.056 | 15806 |
| *Brycon falcatus* | 53437 | 91 | 42.60% | 0.048 | 16848 |
| *Brycon falcatus* | 53445 | 184 | 42.58% | 0.051 | 16838 |
| *Brycon henni* | NC026873.1 | NA | 44.57% | 0.070 | 16885 |
| *Brycon meeki* | Bme008 | 23 | 44.30% | 0.057 | 16967 |
| *Brycon moorei* | C19274 | 187 | 42.38% | 0.058 | 16840 |
| *Brycon nattereri* | NC051927.1 | NA | 41.96% | 0.053 | 16837 |
| *Brycon oligolepis* | Bol019 | 61 | 42.46% | 0.057 | 16884 |
| *Brycon orbignyanus* | Voucher120457 | 1230 | 42.71% | 0.061 | 16877 |
| *Brycon orbignyanus* | NC024938.1 | NA | 44.27% | 0.061 | 16800 |
| *Brycon rubricauda* | A183 | 460 | 44.53% | 0.071 | 16871 |
| *Chilobrycon deuterodon* | 45001 | 134 | 43.85% | 0.064 | 16873 |
| *Salminus affinis* | C8284 | 130 | 43.50% | 0.015 | 17769 |
| *Salminus brasiliensis* | NC024941.1 | NA | 44.26% | 0.028 | 17721 |
| *Salminus brasiliensis* | SZAIPI037396-87 | 2480 | 43.68% | 0.029 | 18169 |
| *Salminus brasiliensis* | 21905 | 57 | 43.90% | 0.028 | 17828 |

**Table 3. Gene sizes of Dloop, rRNA and tRNA genes in 19 mitogenomes of Bryconidae species.**

| Species | D_loop | mt-rnr2 | mt-rnr1 | trnD | trnP | trnW | trnC | trnG | trnF | trnL | trnM | trnY | trnK | trnR | trnH | trnS | trnL | trnT |
|---|---|---|---|---|---|---|---|---|---|---|---|---|---|---|---|---|---|---|
| *B. meeki* Bme008 | 1285 | 1687 | 952 | 69 | 73 | 72 | 66 | 71 | 68 | 75 | 69 | 71 | 76 | 70 | 69 | 68 | 73 | 72 |
| *B. chagrensis* 28B05 | 1256 | 1687 | 953 | 69 | 73 | 72 | 66 | 71 | 68 | 75 | 69 | 71 | 76 | 70 | 69 | 68 | 73 | 72 |
| *B. oligolepis* Bol019 | 1208 | 1681 | 952 | 73 | 70 | 71 | 66 | 72 | 68 | 75 | 69 | 71 | 75 | 70 | 69 | 68 | 73 | 72 |
| *B. henni* NC026873.1 | 1207 | 1678 | 953 | 73 | 70 | 71 | 66 | 72 | 68 | 75 | 69 | 71 | 76 | 70 | 69 | 68 | 73 | 72 |
| *B. rubricauda* A183 | 1192 | 1685 | 957 | 73 | 70 | 72 | 66 | 72 | 69 | 75 | 70 | 71 | 75 | 70 | 70 | 68 | 73 | 73 |
| *Ch. deuterodon* 45001 | 1113 | 1683 | 957 | 73 | 69 | 71 | 66 | 72 | 68 | 75 | 69 | 71 | 76 | 70 | 69 | 68 | 73 | 73 |
| *B. orbignyanus* SRR6243207 | 1236 | 1675 | 952 | 73 | 69 | 71 | 66 | 72 | 68 | 75 | 70 | 71 | 76 | 70 | 69 | 68 | 73 | 72 |
| *B. orbignyanus* NC024938.1 | 1159 | 1675 | 952 | 73 | 69 | 71 | 66 | 72 | 68 | 75 | 70 | 71 | 76 | 70 | 69 | 68 | 73 | 72 |
| *B. moorei* C19274 | 1206 | 1674 | 952 | 73 | 69 | 71 | 66 | 72 | 68 | 75 | 70 | 71 | 76 | 70 | 69 | 68 | 73 | 72 |
| *B. nattereri* NC051927.1 | 1197 | 1672 | 955 | 73 | 69 | 69 | 68 | 72 | 68 | 75 | 70 | 71 | 76 | 70 | 70 | 68 | 73 | 72 |
| *B. falcatus* 53437 | 1133 | 1672 | 953 | 73 | 69 | 69 | 66 | 72 | 68 | 75 | 70 | 71 | 76 | 70 | 70 | 68 | 73 | 72 |
| *B. falcatus* 53445 | 1123 | 1672 | 953 | 73 | 69 | 69 | 66 | 72 | 68 | 75 | 70 | 71 | 76 | 70 | 70 | 68 | 73 | 72 |
| *B. amazonicus* 58483 | 1119 | 1672 | 953 | 73 | 69 | 69 | 66 | 72 | 68 | 75 | 70 | 71 | 76 | 70 | 69 | 68 | 73 | 72 |
| *B. amazonicus* LBP-14082 | 1099 | 1672 | 953 | 73 | 69 | 69 | 66 | 72 | 68 | 75 | 70 | 71 | 76 | 70 | 69 | 68 | 73 | 72 |
| *S. brasiliensis* SZAIPI037396-87 | 2574 | 1675 | 947 | 73 | 69 | 71 | 66 | 70 | 68 | 74 | 69 | 70 | 76 | 69 | 69 | 67 | 72 | 72 |
| *S. brasiliensis* 21905 | 2232 | 1675 | 947 | 73 | 69 | 71 | 67 | 70 | 68 | 74 | 69 | 70 | 76 | 69 | 69 | 67 | 72 | 72 |
| *S. brasiliensis* NC024941.1 | 2128 | 1675 | 947 | 73 | 69 | 70 | 66 | 70 | 68 | 74 | 69 | 70 | 76 | 69 | 69 | 67 | 72 | 72 |
| *S. affinis* C8284 | 2179 | 1674 | 946 | 73 | 69 | 70 | 66 | 70 | 68 | 74 | 69 | 70 | 76 | 69 | 69 | 67 | 72 | 72 |

Seven tRNA genes had the same length in all studied species (*trn*N: 73 bp, *trn*V: 72 bp, *trn*I: 72 bp, *trn*Q: 71 bp, *trn*S: 71, *trn*A: 69 bp, *trn*E: 69 bp). The samples from northwestern South America are shaded in gray.

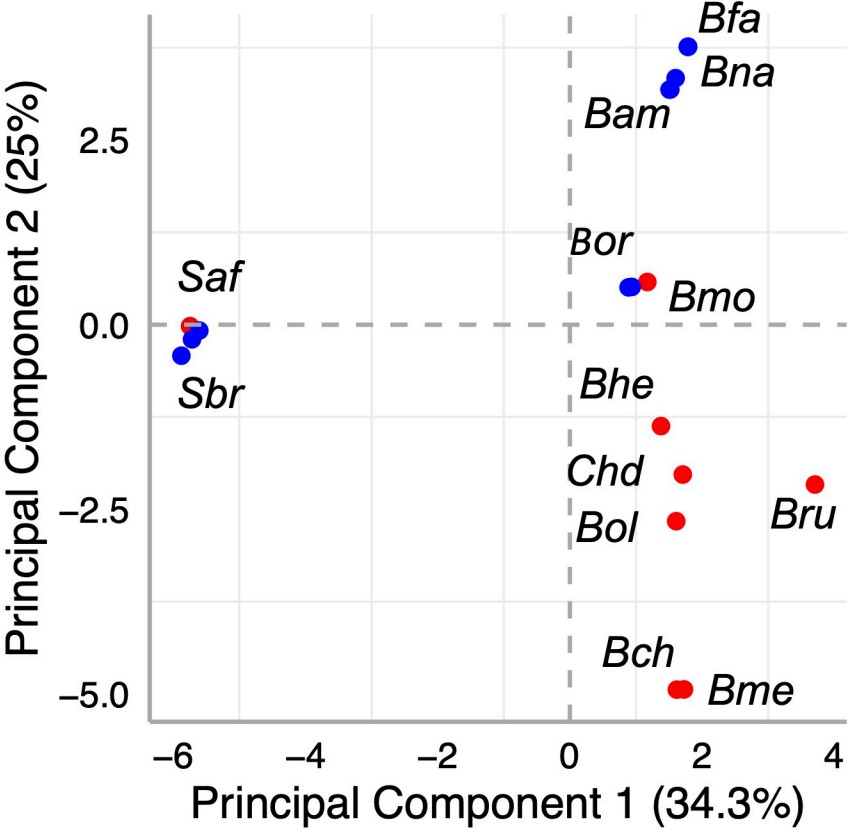

**Fig 3. Plot of the two first principal components based on the length variations of Coding Sequences (CDSs), ribosomal RNA (rRNA), transfer RNA (tRNA), and the D-loop region of Bryconidae mitogenomes.** Cis- and trans-Andean species are denoted in blue and red, respectively. Bam: *Brycon amazonicus*; Bch: *Brycon chagrensis*; Bfa: *Brycon falcatus*; Bhe: *Brycon henni*; Bme: *Brycon meeki*; Bmo: *Brycon moorei*; Bna: *Brycon nattereri*; Bol: *Brycon oligolepis*; Bor: *Brycon orbignyanus*; Bru: *Brycon rubricauda*; Chd: *Chilobrycon deuterodon*; Saf: *Salminus affinis*; Sbr: *Salminus brasiliensis*.

Table 3). In contrast, the variations in *mt-rnr1* gene (946–992 bp; Median: 953 bp) and *mt-rnr2* gene (1,672–1,687 bp; Median 1,675 bp) were stable within the species. Minor differences in lengths (1–4 bp; Table 3) were observed in 15 of 22 tRNAs genes, whereas seven had the same length in all studied species (*trn*N: 73 bp, *trn*V: 72 bp, *trn*I: 72 bp, *trn*Q: 71 bp, *trn*S: 71, *trn*A: 69 bp, *trn*E: 69 bp).

In the coding protein genes (Table 4), the greatest differences in length were observed in the CDSs of the genes *nd6* (480–519 pb), *nd5* (1,848–1,863 pb), *cytb* (1,141 bp, slightly longer with 1,150 bp for *B. meeki* and *B. chagrensis*), *cox1* (1,557 bp, slightly shorter with 1,548 bp in *B. amazonicus*, *B. henni* and *B. nattereri*), and *atp6* (683 bp., slightly shorter with 677 bp for *B. moorei*). Minor variations were found in the length of the CDSs of the genes *nd1* (975 bp, one extra codon is annotated in *Brycon nattereri*, *B. amazonicus*, *B. falcatus*, *B. orbignyanus*, and *B. moorei*), *atp8* (165 bp, with one extra codon annotated for *B. rubricauda*, *B. nattereri*, *B. amazonicus*, and *B. falcatus*), *nd4l* (294–297 bp), *nd4* (1384 bp in *B. amazonicus* and *B. falcatus*, with one triplet shorter in the other species), *nd3* (349 bp in the trans-Andean species, with one triplet shorter in the cis-Andean species), *nd2* (1,045–1,046 bp; the partial stop codon is completed by the addition of A residues at the 3' end of the mRNA, doi:10.1093/gbe/evw195), and *cox2* (691–692 bp). Only one of 13 coding protein genes exhibited the same length in all studied species (*cox3*: 784 bp).

**Table 4. Gene sizes of protein coding genes in19 mitogenomes of 13 Bryconidae species.**

| Species | nd6 | nd5 | cytb | cox1 | atp6 | nd1 | atp8 | nd3 | nd4l | nd4 | nd2 | cox2 |
|---|---|---|---|---|---|---|---|---|---|---|---|---|
| *B. meeki* Bme008 | 519 | 1848 | 1150 | 1557 | 683 | 975 | 165 | 349 | 297 | 1381 | 1046 | 691 |
| *B. chagrensis* 28B05 | 519 | 1848 | 1150 | 1557 | 683 | 975 | 165 | 349 | 297 | 1381 | 1046 | 691 |
| *B. oligolepis* Bol019 | 519 | 1851 | 1141 | 1557 | 683 | 975 | 165 | 349 | 297 | 1381 | 1046 | 692 |
| *B. henni* NC026873.1 | 519 | 1851 | 1141 | 1548 | 683 | 975 | 165 | 349 | 297 | 1381 | 1046 | 691 |
| *B. rubricauda* A183 | 519 | 1848 | 1141 | 1557 | 683 | 975 | 168 | 349 | 297 | 1381 | 1046 | 691 |
| *Ch. deuterodon* 45001 | 480 | 1851 | 1141 | 1557 | 683 | 975 | 165 | 349 | 297 | 1381 | 1046 | 691 |
| *B. orbignyanus* SRR6243207 | 513 | 1857 | 1141 | 1557 | 683 | 978 | 165 | 346 | 297 | 1381 | 1046 | 691 |
| *B. orbignyanus* NC024938.1 | 513 | 1857 | 1141 | 1557 | 683 | 978 | 165 | 346 | 297 | 1381 | 1046 | 691 |
| *B. moorei* C19274 | 513 | 1854 | 1141 | 1557 | 677 | 978 | 165 | 346 | 297 | 1381 | 1046 | 691 |
| *B. nattereri* NC051927.1 | 513 | 1854 | 1141 | 1548 | 683 | 978 | 168 | 346 | 297 | 1381 | 1045 | 691 |
| *B. falcatus* 53437 | 513 | 1854 | 1141 | 1557 | 683 | 978 | 168 | 346 | 294 | 1384 | 1045 | 691 |
| *B. falcatus* 53445 | 513 | 1854 | 1141 | 1557 | 683 | 978 | 168 | 346 | 294 | 1384 | 1045 | 691 |
| *B. amazonicus* 58483/LBP-14082 | 513 | 1854 | 1141 | 1548 | 683 | 978 | 168 | 346 | 297 | 1384 | 1045 | 691 |
| *B. amazonicus* LBP-14082 | 513 | 1854 | 1141 | 1548 | 683 | 978 | 168 | 346 | 297 | 1384 | 1045 | 691 |
| *S. brasiliensis* SZAIPI037396-87 | 510 | 1863 | 1141 | 1557 | 683 | 975 | 165 | 346 | 297 | 1381 | 1045 | 691 |
| *S. brasiliensis* 21905 | 510 | 1863 | 1141 | 1557 | 683 | 975 | 165 | 346 | 297 | 1381 | 1045 | 691 |
| *S. brasiliensis* NC024941.1 | 510 | 1863 | 1141 | 1557 | 683 | 975 | 165 | 346 | 297 | 1381 | 1045 | 691 |
| *S. affinis* C8284 | 510 | 1863 | 1141 | 1557 | 683 | 975 | 165 | 346 | 297 | 1381 | 1045 | 691 |

The *cox3* gene have 784 bp in all studied species. The samples from northwestern South America are shaded in gray.

The phylogenetic relationships based on the 19 mitochondrial genomes of 13 Bryconidae species, 13 concatenated CDSs and two rRNAs, depict the family as a well-supported monophyletic group (100 UFB support, 100 gene concordance factor -GCF, and 43.3 site concordance factor -SCF; Fig 4). Furthermore, two main sister clades can be observed: 1) trans-Andean *Brycon* (north-western South America) from the Pacific slope + *Brycon chagrensis* + *Chilobrycon*; and 2) cis-Andean *Brycon* + *Salminus* (eastern South America) + congeners from the Magdalena-Cauca Basin, with UFB support values of 100% and 99%, respectively.

## Discussion

This study assembled, compared, and explored the phylogenetic relationships of 15 mitogenomes corresponding to 11 species of Bryconidae. This included the de novo sequencing of mitochondrial genomes for five species from north-western South America, as well as the assembly of 10 mitochondrial genomes from NGS genomic or transcriptomic data obtained from the SRA database. All the expected protein-coding, rRNA, and tRNA genes were annotated in the Bryconidae mitogenomes. However, in *B. chagrensis* (29B07) it was not possible to detect the D-loop region and a tRNA in the contig assembled from RNA-seq data.

The mitogenome annotation results indicate the same mitochondrial genome structure and synteny among Bryconidae and other Characiformes mitogenomes [21, 22, 24, 34–38]. The overall AT skews (0.015–0.071; Median: 0.057) indicate a greater AT bias compared to that previously observed in Characidae [36–40].

*Salminus* mitogenomes were the largest followed by Bryconidae from northwestern and southeastern South America. Differences in length are mainly explained by variations among the D-loop regions, which exhibited variations even in mitogenomes from the same species as observed in *S. brasiliensis*, *B. orbignyanus*, and *B. amazonicus*. These D-loop region lengths are also larger than in other fish species that exhibited ranges from 724 to 1,401 nts [37, 41–44].

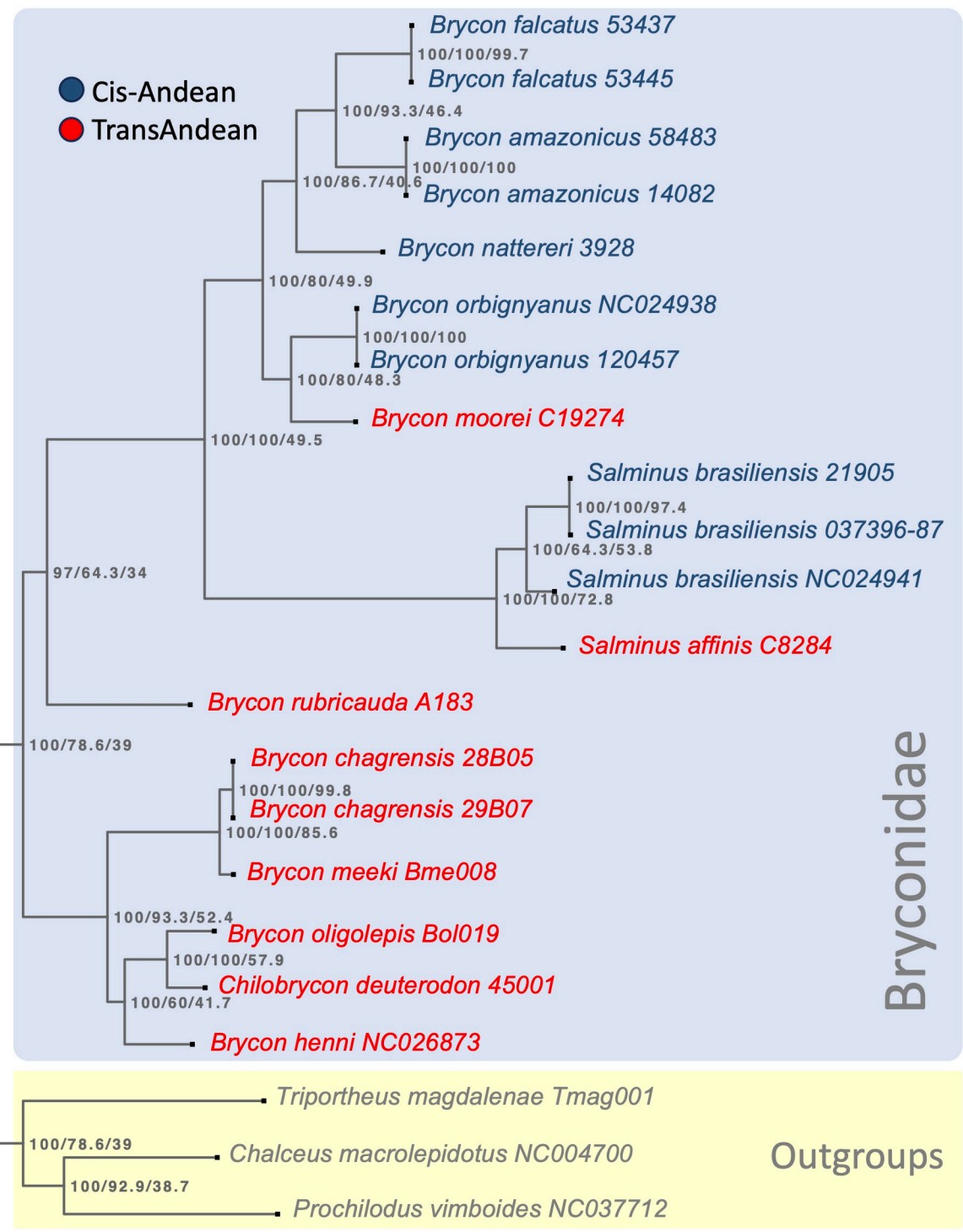

**Fig 4.** Mitochondrial Genome Evolution in Bryconidae: A Maximum Likelihood Phylogenetic Tree was constructed from the alignment of 13 protein-coding sequences (CDS) and 2 mitochondrial ribosomal RNAs (rRNAs) across 19 mitogenomes of 13 species within the Bryconidae. The tree is rooted using *Triportheus magdalenae*, *Chalceus macrolepidotus*, and *Prochilodus vimboides* as outgroups and is annotated with supports indicated by concordance factors (UFB/gcf/scf), specifically using UFB 5000. Species from the cis-Andean region are color-coded in blue, while those from the trans-Andean region are highlighted in red.

Differences in the control region length explain variations in mitochondrial genome size in most vertebrates [45–47] and copy number variations have also been found even within the same individuals [42].

The extension of the *mt-rnr1* and *mt-rnr2* genes showed variations typical of the species. Both genes have been demonstrated to exhibit regions with large variation in length and sequence [44]. An additional variation source was observed in 15 tRNAs and 12 coding protein genes, specially *nd6*, *nd5*, *cytb*, *cox1*, and *atp6*. The tRNA size variation has been documented for D-loop, T-loop, V-loop and even D stem [44]. Additionally, high gene length variation levels were also found in *nd5*, *cytb*, and *cox1* in other fish species, and was attributed to both gene size and gene rearrangement [44]. However, contrasting those results, *nd2* in Bryconidae does not show high size variation levels, whereas *nd5* and *atp6* showed high length variation levels, suggesting that these patterns may be related to the evolutionary trajectories of the taxa. As shown by the PCA, the mitogenomes size variation pattern is highly congruent with their phylogenetic relationships [14, this study] indicating that these differences represent an evolutionary signal. Moreover, the mitogenome size variation pattern showed the closer relationship in the mitogenomes size variation pattern among *B. moorei* and southeastern South America, which is consistent with their closer phylogenetic relationships.

Since only 19 mitogenomes corresponding to 13 species of Bryconidae were included in our analysis, the phylogenetic relationships found may be influenced by an incomplete sampling. Despite this limitation, the current analysis recovers the phylogenetic relationships previously reported and offers new insights into the origin and diversification of Bryconidae groups. The trans-Andean species from the Pacific slope drainages were grouped, with *B. meeki* and *B. chagrensis* as sister species (UFB: 100, GCF: 100; SCF: 85.3), while *B. oligolepis* was clustered with *Chilobrycon* (UFB: 100, GCF: 100, SCF 57.6). *Brycon henni* appears as an ancestral lineage in this last clade, also with 100% UFB support. Previously described phylogenetic relationships of *B. chagrensis* as the sister clade of *Chilobrycon* and *B. henni* [14] indicate that *B. chagrensis* + *B. meeki* is also a sister clade of (*B. henni* + (*Chilobrycon* + *B. oligolepis*)).

On the other hand, in the cis-Andean + Magdalena-Cauca Basin (*Brycon moorei* and *Salminus affinis*) clade, the remaining eight species were clustered, displaying 100% UFB support, with 100% and 49.5% concordance factors for genes and sites, respectively. It is noteworthy that *B. rubricauda* was positioned as the basal lineage of the cis-Andean clade. Furthermore, two main lineages were formed within this clade: one encompasses the *Salminus* species (UFB 100, GCF 100, SCF 72.8), and the other comprises the species *B. orbignyanus*, *B. moorei*, *B. nattereri*, *B. falcatus*, and *B. amazonicus* (with UFB 100, GCF 80.0, SCF 49.9).

Furthermore, *B. moorei* was found to be clustered with *B. orbignyanus* (UFB: 100, GCF: 80; SCF: 48), and *S. affinis* was grouped with *S. brasiliensis* (UFB: 100, GCF: 100; SCF: 72.9). This agrees with previous report based on *cytb* and *cox1* [14], according to which *B. moorei* is phylogenetically related to the cis-Andean congeners closely related to *Salminus*. Several other authors have also previously proposed the close relationships among *Brycon* and *Salminus* [14, 20, 48–51]. Despite this close phylogenetic relationship, *Salminus* can be unequivocally recognized using alternatively morphological [18] and molecular characters [14, 52, this study].

This mitochondrial phylogenomic analysis supports the monophyly of Bryconidae and *Salminus*. However, as reported by other authors, current members of *Brycon* are split in separated lineages [14, 49]. Interestingly, members from Pacific slopes drainages are more closely related to the monotypic *Chilobrycon* than the remaining congeners, corroborating the need to revise the taxonomy of trans-Andean *Brycon*. A plausible alternative would be to expand the current concept of *Chilobrycon* to include *B. chagrensis*, *B. henni*, *B. meeki* and *B. oligolepis*, but this action requires a further comprehensive taxonomic and morphological revision.

Based on the similar topology with a previous study [14], one general hypothesis that can be drawn based on this mitochondrial phylogenomic analysis is that the diversification of the ancestor of Bryconidae originated in north-western South America, followed by vicariant events that isolated the Pacific clade, which subsequently invaded Central America. The hypothesis suggesting a potential invasion of Central America by Bryconidae, as previously proposed [14], aligns well with the notion of a stepwise colonization of *Hyphessobrycon* from the Pacific slope of northwestern South America to middle America [53]. This reinforces the need for continued investigation and exploration to refine the historical biogeography and evolutionary dynamics understanding within the Bryconidae family.

The expansion of Bryconidae in South America was proposed by Abe et al. (2014) [14] based on the hypothesis of López-Fernández and Alberts (2011) [54] according to which substantial marine regressions in the Oligocene, akin to earlier periods, revealed extensive interior floodplains, a scenario that is believed to have expedited the rapid expansion of freshwater habitats. The common ancestor of the clade that includes *B. rubricauda*, *Salminus*, and the remaining *Brycon* species suggests that the expansion occurred from northwestern towards the eastern and southern South America. This hypothetical scenario should be examined in future biogeographic studies using mitochondrial and nuclear markers with a wider taxonomic and geographic representation.

In conclusion, prior to this study, only four mitogenomes were available for 52 Bryconidae species. This study, in addition to shedding new light on mitogenomic characteristics and evolutionary trajectories among Bryconidae fishes and providing a valuable resource for environmental DNA applications, molecular ecology, and phylogenetics, provided 15 additional mitogenomes, for a total of 19 mitogenomes corresponding to 13 species. Despite the latter, the inclusion of further mitogenomes and the examination of multiple nuclear loci within this family are imperative for a holistic understanding of their diversity and evolutionary panorama.

## Acknowledgments

The authors would like to thank Colección de Ictiología—Universidad de Antioquia, as well as Universidad de Córdoba for kindly providing the samples used in this study.

## Author Contributions

**Conceptualization:** Edna Judith Márquez, Juan Fernando Alzate.

**Data curation:** Edna Judith Márquez, Daniel Alfredo Gómez-Chavarría, Juan Fernando Alzate.

**Formal analysis:** Edna Judith Márquez, Daniel Alfredo Gómez-Chavarría, Juan Fernando Alzate.

**Funding acquisition:** Edna Judith Márquez.

**Investigation:** Edna Judith Márquez, Juan Fernando Alzate.

**Methodology:** Edna Judith Márquez, Daniel Alfredo Gómez-Chavarría, Juan Fernando Alzate.

**Project administration:** Edna Judith Márquez.

**Resources:** Edna Judith Márquez.

**Software:** Edna Judith Márquez, Daniel Alfredo Gómez-Chavarría, Juan Fernando Alzate.

**Supervision:** Edna Judith Márquez, Daniel Alfredo Gómez-Chavarría, Juan Fernando Alzate.

**Validation:** Edna Judith Márquez, Daniel Alfredo Gómez-Chavarría, Juan Fernando Alzate.

**Visualization:** Edna Judith Márquez, Daniel Alfredo Gómez-Chavarría, Juan Fernando Alzate.

**Writing – original draft:** Edna Judith Márquez, Daniel Alfredo Gómez-Chavarría, Juan Fernando Alzate.

**Writing – review & editing:** Edna Judith Márquez, Daniel Alfredo Gómez-Chavarría, Juan Fernando Alzate.

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
