## [Decision Letter · Decision Letter 0]

15 Apr 2024

PONE-D-24-08613Exploring the mitochondrial genomes and phylogenetic relationships of eleven Bryconidae speciesPLOS ONE

Dear Dr. Márquez,

Thank you for submitting your manuscript to PLOS ONE. After careful consideration, we feel that it has merit but does not fully meet PLOS ONE’s publication criteria as it currently stands. Therefore, we invite you to submit a revised version of the manuscript that addresses the points raised during the review process. Please see my own comments an the reviews of two reviewers below.

We look forward to receiving your revised manuscript.

Kind regards,

Roberto E. Reis, Ph.D.

Academic Editor

PLOS ONE

Journal Requirements:

Additional Editor Comments:

The manuscript is mostly well written, but its title and abstract fits better in a paper on genetic data than in a phylogenetic contribution. The authors have somewhat mixed concepts and failed to explore and extract the full potential of the data so that this manuscript could be considered a phylogenetic contribution. Both reviewers mentioned problems related to be above and presented excellent sugestions, which I agree with. Further on these general issues, I have a few format corrections:

1. Only cite species authorities (author, year) the first time a name appears in the manuscript.

2. Chose and standradize the language the geographic descriptors. You use both Spanish (quebrada, río) and english (River).

Reviewers' comments:

Reviewer's Responses to Questions

**Comments to the Author**

1. Is the manuscript technically sound, and do the data support the conclusions?

Reviewer #1: Yes

Reviewer #2: Yes

2. Has the statistical analysis been performed appropriately and rigorously? 

Reviewer #1: I Don't Know

Reviewer #2: Yes

3. Have the authors made all data underlying the findings in their manuscript fully available?

Reviewer #1: Yes

Reviewer #2: Yes

4. Is the manuscript presented in an intelligible fashion and written in standard English?

Reviewer #1: Yes

Reviewer #2: Yes

5. Review Comments to the Author

Reviewer #1: Comments on MS PONE-D-24-08613 “Exploring the mitochondrial genomes and phylogenetic relationships of eleven Bryconidae species” by Edna Judith Marquez, Daniel Alfredo Gómez-Chavarria & Juan Fernando Alzate.

That’s an interesting contribution that improves our understanding of the phylogenetic relationships within the family Bryconidae, especially because it focuses transandean taxa, that have been neglected in previous phylogenies of the family. I consider that the manuscript has a lot of merit and should be published, but I have a few suggestions for its improvement, that follow below.

Title: I suggest changing “eleven” for “Transandean” to emphasize what constitutes to be the main contribution of the present MS regarding the knowledge of Bryconidae.

Abstract, l. 31-33: It would be preferrable to treat the genus Salminus as another clade as this genus has been only considered as closely related to Brycon after phylogenies using molecular data have demonstrated it. The fact that Salminus is a cis/transandean lineage is known since the description of Salminus affinis by Steindachner in 1877, as there was never any doubt that Salminus is monophyletic. It would be better simply to state that Brycon rubricauda is sister to a clade including Salminus and Brycon species mostly from cis-andean systems but including a species from the Magdalena/Cauca, B. moorei. Notice also that Brycon henni and a Brycon of the B. atrocaudatus species complex occur at the Magdalena/Cauca basin, so the generalization that Bryconidae from the Magdalena/Cauca group with cisandean taxa is incorrect.

Introduction, P. 3, l. 55-56. There are actually 44 Brycon species currently considered as valid. The authors should cite the much more updated and broad revision of the cisandean Brycon by Lima (2017) instead of Lima (2004) for that information, since the latter reference actually mentions 41 valid species of Brycon, which was the number of species recognized at that time, and it is only a species description, not a revision.

P.3, L. 56-57: Lima (2004) does not state at any point that Brycon shows its greater diversity at transandean rivers. The authors should instead cite Lima (2017), which mentions at p. 4 that half of the Brycon species currently recognized as valid are transandean, so the diversity of the genus is equally divided between the transandean and cisandean South America. It is probably true that the area between Panama and coastal northern Peru concentrates the largest diversity in the family per area, the problem is that this need to be clearly demonstrated, but it hasn’t so far and at the moment there are no references that the authors could cite to support that assertion.

Material and methods, L. 83-93: Not sure that the authors need to go into this detail in describing their samples. A table as an additional file would suffice for that.

P. 14, L. 305-314: The authors does not address the fact that their results are affected by the obvious fact that they are just sampling a small fraction of the diversity of the family, and much less terminals than the largest published phylogenetic study of the family (Abe et al., 2014). They should be more cautious when inferring relationships and remind the readers that difference in results may be simply a result of sampling bias. For example, there is very compelling morphological evidence that Brycon henni forms a monophyletic clade with B. stolzmanni and B. coxeyi (see Lima, 2017), two species that have not being analysed in the present study (nor in Abe et al., 2014). There is no need for recommending a “taxonomic revision of the group” as it is well-know that the taxonomy of transandean Brycon is in a state of disarray, so the authors are making an unnecessary recommendation.

P. 15, l. 326-329: Salminus affinis is only recovered as sister of S. brasiliensis because the authors have not sampled any other Salminus species! See Abe et al. (2104) and Machado et al. (2018) (which should be cited by the authors), and check the discussion on the phylogeny of Salminus by Lima (2022). As noticed above, the authors are mistakenly assuming relationships without considering that their study, despite its merits in including taxa absent from other phylogenetic analyses of the family, it nonetheless only includes a small proportion of the diversity of the family and consequently, much caution should be used in inferring relationships.

P. 15, L. 338-345: The authors should not use the word “migration” which has a very different biological sense, and use instead “expansion” as Abe et al. (2014) did.

References (only the ones not cited in the MS)

Lima, F.C.T. 2017. A revision of the cis-andean species of the genus Brycon Müller & Troschel (Characiformes: Characidae). Zootaxa, 4222(1): 1-189

https://doi.org/10.11646/zootaxa.4222.1.1

Machado, C.B., Galetti Jr., P.M. & Carnaval, A.C. (2018) Bayesian analyses detect a history of both vicariance and geodispersal

in Neotropical freshwater fishes. Journal of Biogeography, 45, 1313–1325.

https://doi.org/10.1111/jbi.13207

Reviewer #2: I Congratulate the authors on the interesting investigation on such important fishes. Overall the manuscript is well-written, but need clarification on several aspects detailed below.

The main one involves the number of sampled species on the paper. Along most of the manuscript 11 species are mentioned, but the figures and the tables show 13, and 19 species are mentioned at least once on the text. This must be thoroughly reviewed.

On the scientific aspects of the paper, most of my criticism is directed towards the Phylogenetic portions of the manuscript. Not only the presentation of systematic, taxonomic, and biogeographic interpretations of the results obtained seem to be confusing, but also because the phylogenetic analyses and their developments correspond to a secondary portion of the paper that the entire proposal of phylogenetic investigation seem misleading. On the other hand, I do agree with the authors’ intention perform a comparative/phylogenetic evaluation of the Mitogenomes obtained. Therefore, to reduce any bias presented in the title and objectives of the paper, I would suggest changing the title of tha manuscript to “Comparative evaluation of the mitochondrial genomes of thirteen members of the Bryconidae (Ostariophysi; Characiformes)”, which would represent more accurately the aspects of the paper.

In addition, I have found instances where the text has some sort of problems:

Lines

2 - Authors should include taxonomic indexers after Bryconidae, to locate readers from other fields on the branch of the tree of life the paper is aiming.

21-22 - A phylogeny including 11 among 50 species is far from bridging a gap, especially if a previous study has presented a hypothesis with more than half the diversity of the species of the group which is barely explored in the manuscript

23 - the number of species comprised in the study is confusing, the authors mention 11 here and elsewhere in the text, but the tables show 13 species instead. this information should be reviewed and if the numbers are correct, clear sentences on the result section explaining which information or analyses apply to 11 or 13 species.

33 - The term “lack of monophyly” seems innappropriate as monophyly is not “present or absent” in a group. Instead, a group is monophyletic or not.

189 - table 2 caption mentions 19 species, when 13 are presented.

251-252 - see comments above.

282-283 - this sentence is confusing, as it seems the authors are comparing the mitogenomes of Salminus and Bryconids with other groups among the examined, which is not accurate. I suggest rewording.

321-322 UFB and Bootstrap are used as synonyms, but UFB would fit better the methods presented in the paper.

326-328 this sentence should be merged with the previous one, concluding that Salminus is actually a species of Brycon with several modifications.

329 change “monophyletic grouping of Bryconidae” to “monophyly of Bryconidae”. At the same time, with two outgroups, that is not actually at the reach of the current results.

329-337 this entire paragraphs lacks logical or theoretical cohesion: the “validity of trans-andean Brycon” has absolutely no connection with their close relationship with Chilobrycon or the fact that they occur in the pacific slope. Are the authors suggesting that those species need to be accommodated in Chilobrycon as well? Although I agree with the suggestion, this discussion is taxonomic, being mixed with biogeographical inferences along the rest of the paragraph, and it should be reorganized in distinct portions of the text.

346 - add article “by the Bryconidae”

350-351 “the Bryconidae family” is a redundance as all taxa ending in -idae are in the family rank.

6. PLOS authors have the option to publish the peer review history of their article (what does this mean?). If published, this will include your full peer review and any attached files.

Reviewer #1: No

Reviewer #2: **Yes: **Andre Luiz Netto-Ferreira

---

## [Author Response · Author response to Decision Letter 0]

30 May 2024

Dear Editor 

We greatly appreciate the detailed and constructive feedback from the reviewers. We have revised the manuscript according to their valuable recommendations, resulting in significant improvements to our paper.

We hope that the manuscript is now suitable for publication in Plos One.

PONE-D-24-08613

Exploring the mitochondrial genomes and phylogenetic relationships of eleven Bryconidae species

PLOS ONE

Dear Dr. Márquez,

Thank you for submitting your manuscript to PLOS ONE. After careful consideration, we feel that it has merit but does not fully meet PLOS ONE’s publication criteria as it currently stands. Therefore, we invite you to submit a revised version of the manuscript that addresses the points raised during the review process. Please see my own comments an the reviews of two reviewers below.

We look forward to receiving your revised manuscript.

Kind regards,

Roberto E. Reis, Ph.D.

Academic Editor

PLOS ONE

Journal Requirements:

Additional Editor Comments:

The manuscript is mostly well written, but its title and abstract fits better in a paper on genetic data than in a phylogenetic contribution. The authors have somewhat mixed concepts and failed to explore and extract the full potential of the data so that this manuscript could be considered a phylogenetic contribution. Both reviewers mentioned problems related to be above and presented excellent sugestions, which I agree with.

We have edited these sections as described below.

Further on these general issues, I have a few format corrections:

1. Only cite species authorities (author, year) the first time a name appears in the manuscript.

Done.

2. Chose and standradize the language the geographic descriptors. You use both Spanish (quebrada, río) and english (River).

As suggested by Reviewer #1, we have removed this information.

Reviewers' comments:

Reviewer's Responses to Questions

Comments to the Author

1. Is the manuscript technically sound, and do the data support the conclusions?

Reviewer #1: Yes

Reviewer #2: Yes

2. Has the statistical analysis been performed appropriately and rigorously?

Reviewer #1: I Don't Know

Reviewer #2: Yes

3. Have the authors made all data underlying the findings in their manuscript fully available?

Reviewer #1: Yes

Reviewer #2: Yes

4. Is the manuscript presented in an intelligible fashion and written in standard English?

Reviewer #1: Yes

Reviewer #2: Yes

5. Review Comments to the Author

Reviewer #1: Comments on MS PONE-D-24-08613 “Exploring the mitochondrial genomes and phylogenetic relationships of eleven Bryconidae species” by Edna Judith Marquez, Daniel Alfredo Gómez-Chavarria & Juan Fernando Alzate.

That’s an interesting contribution that improves our understanding of the phylogenetic relationships within the family Bryconidae, especially because it focuses transandean taxa, that have been neglected in previous phylogenies of the family. I consider that the manuscript has a lot of merit and should be published, but I have a few suggestions for its improvement, that follow below.

Title: I suggest changing “eleven” for “Transandean” to emphasize what constitutes to be the main contribution of the present MS regarding the knowledge of Bryconidae.

Done. Now, considering comments of both reviewers: “Exploring the mitochondrial genomes and phylogenetic relationships of trans-Andean Bryconidae species (Actinopterygii: Ostariophysi: Characiformes)”

Abstract, l. 31-33: It would be preferrable to treat the genus Salminus as another clade as this genus has been only considered as closely related to Brycon after phylogenies using molecular data have demonstrated it. The fact that Salminus is a cis/transandean lineage is known since the description of Salminus affinis by Steindachner in 1877, as there was never any doubt that Salminus is monophyletic. It would be better simply to state that Brycon rubricauda is sister to a clade including Salminus and Brycon species mostly from cis-andean systems but including a species from the Magdalena/Cauca, B. moorei. Notice also that Brycon henni and a Brycon of the B. atrocaudatus species complex occur at the Magdalena/Cauca basin, so the generalization that Bryconidae from the Magdalena/Cauca group with cisandean taxa is incorrect.

Done. Now: “The other clade grouped the trans-Andean species from the Magdalena-Cauca Basin Brycon moorei and Salminus affinis, with their respective cis-Andean congeners (found in eastern South America), with Brycon rubricauda as its sister clade”

In Discussion: “On the other hand, in the cis-Andean + Magdalena-Cauca Basin (Brycon moorei and Salminus affinis) clade, the remaining eight species were clustered, displaying 100% UFB support, with 100% and 49.5% concordance factors for genes and sites, respectively.”

Introduction, P. 3, l. 55-56. There are actually 44 Brycon species currently considered as valid. The authors should cite the much more updated and broad revision of the cisandean Brycon by Lima (2017) instead of Lima (2004) for that information, since the latter reference actually mentions 41 valid species of Brycon, which was the number of species recognized at that time, and it is only a species description, not a revision.

Done

P.3, L. 56-57: Lima (2004) does not state at any point that Brycon shows its greater diversity at transandean rivers. The authors should instead cite Lima (2017), which mentions at p. 4 that half of the Brycon species currently recognized as valid are transandean, so the diversity of the genus is equally divided between the transandean and cisandean South America. It is probably true that the area between Panama and coastal northern Peru concentrates the largest diversity in the family per area, the problem is that this need to be clearly demonstrated, but it hasn’t so far and at the moment there are no references that the authors could cite to support that assertion.

Done. We removed “showing its greater taxonomic diversity in Panama and the Colombian and Ecuadorian trans-Andean rivers [15]”

Material and methods, L. 83-93: Not sure that the authors need to go into this detail in describing their samples. A table as an additional file would suffice for that.

P. 14, L. 305-314: The authors does not address the fact that their results are affected by the obvious fact that they are just sampling a small fraction of the diversity of the family, and much less terminals than the largest published phylogenetic study of the family (Abe et al., 2014). They should be more cautious when inferring relationships and remind the readers that difference in results may be simply a result of sampling bias. For example, there is very compelling morphological evidence that Brycon henni forms a monophyletic clade with B. stolzmanni and B. coxeyi (see Lima, 2017), two species that have not being analysed in the present study (nor in Abe et al., 2014). There is no need for recommending a “taxonomic revision of the group” as it is well-know that the taxonomy of transandean Brycon is in a state of disarray, so the authors are making an unnecessary recommendation.

Now: “Since only 19 mitogenomes corresponding to 13 species of Bryconidae were included in our analysis, the phylogenetic relationships found may be influenced by an incomplete sampling. Despite this limitation, the current analysis recovers the phylogenetic relationships previously reported and offers new insights into the origin and diversification of Bryconidae groups. The trans-Andean species from the Pacific slope drainages were grouped, with B. meeki and B. chagrensis as sister species (UFB: 100, GCF: 100; SCF: 85.3), while B. oligolepis was clustered with Chilobrycon (UFB: 100, GCF: 100, SCF 57.6). Brycon henni appears as an ancestral lineage in this last clade, also with 100% UFB support. Previously described phylogenetic relationships of B. chagrensis as the sister clade of Chilobrycon and B. henni [14] indicate that B. chagrensis + B. meeki is also a sister clade of (B. henni + (Chilobrycon + B. oligolepis)).

P. 15, l. 326-329: Salminus affinis is only recovered as sister of S. brasiliensis because the authors have not sampled any other Salminus species! See Abe et al. (2104) and Machado et al. (2018) (which should be cited by the authors), and check the discussion on the phylogeny of Salminus by Lima (2022). As noticed above, the authors are mistakenly assuming relationships without considering that their study, despite its merits in including taxa absent from other phylogenetic analyses of the family, it nonetheless only includes a small proportion of the diversity of the family and consequently, much caution should be used in inferring relationships.

We removed “and S. affinis is a sister species of S. brasiliensis” and include an explanation to clarify this idea. 

Now: “Furthermore, B. moorei was found to be clustered with B. orbignyanus (UFB: 100, GCF: 80; SCF: 48), and S. affinis was grouped with S. brasiliensis (UFB: 100, GCF: 100; SCF: 72.9). This agrees with previous report based on cytb and cox1 [14], according to which B. moorei is phylogenetically related to the cis-Andean congeners closely related to Salminus. Several other authors have also previously proposed the close relationships among Brycon and Salminus [14, 20, 48-51]. Despite this close phylogenetic relationship, Salminus can be unequivocally recognized using alternatively morphological [18] and molecular characters [14, 52, this study]."

P. 15, L. 338-345: The authors should not use the word “migration” which has a very different biological sense, and use instead “expansion” as Abe et al. (2014) did.

We have reordered this section including recommendations from Reviewer #2. Please see below.

References (only the ones not cited in the MS)

Lima, F.C.T. 2017. A revision of the cis-andean species of the genus Brycon Müller & Troschel (Characiformes: Characidae). Zootaxa, 4222(1): 1-189

https://doi.org/10.11646/zootaxa.4222.1.1

Machado, C.B., Galetti Jr., P.M. & Carnaval, A.C. (2018) Bayesian analyses detect a history of both vicariance and geodispersal

in Neotropical freshwater fishes. Journal of Biogeography, 45, 1313–1325.

https://doi.org/10.1111/jbi.13207

Reviewer #2: I Congratulate the authors on the interesting investigation on such important fishes. Overall the manuscript is well-written, but need clarification on several aspects detailed below.

The main one involves the number of sampled species on the paper. Along most of the manuscript 11 species are mentioned, but the figures and the tables show 13, and 19 species are mentioned at least once on the text. This must be thoroughly reviewed.

We assembled 15 mitogenomes from 11 Bryconidae species, but the comparative mitogenomics and evolutionary relationships were analyzed using 19 mitogenomes from 13 species, as we included four previously reported mitogenomes. The title of table 2 was also corrected: “Table 2. Summarized mitogenomic and assemble characteristics of the 13 Bryconidae species.”

On the scientific aspects of the paper, most of my criticism is directed towards the Phylogenetic portions of the manuscript. Not only the presentation of systematic, taxonomic, and biogeographic interpretations of the results obtained seem to be confusing, but also because the phylogenetic analyses and their developments correspond to a secondary portion of the paper that the entire proposal of phylogenetic investigation seem misleading. 

Please, see the edited paragraphs below.

On the other hand, I do agree with the authors’ intention perform a comparative/phylogenetic evaluation of the Mitogenomes obtained. Therefore, to reduce any bias presented in the title and objectives of the paper, I would suggest changing the title of tha manuscript to “Comparative evaluation of the mitochondrial genomes of thirteen members of the Bryconidae (Ostariophysi; Characiformes)”, which would represent more accurately the aspects o

---

## [Decision Letter · Decision Letter 1]

31 Jul 2024

PONE-D-24-08613R1Exploring the mitochondrial genomes and phylogenetic relationships of trans-Andean Bryconidae species (Actinopterygii: Ostariophysi: Characiformes)PLOS ONE

Dear Dr. Márquez,

Thank you for submitting your manuscript to PLOS ONE. After careful consideration, we feel that it has merit but does not fully meet PLOS ONE’s publication criteria as it currently stands. Therefore, we invite you to submit a revised version of the manuscript that addresses the points raised during the review process.

We look forward to receiving your revised manuscript.

Kind regards,

Roberto E. Reis, Ph.D.

Academic Editor

PLOS ONE

Journal Requirements:

**Additional Editor Comments:**

This revised version of the ms in considerably improved compared to its first draft. Please follow the corrections by Reviewer#1 below and resubmit for acceptance.

Reviewers' comments:

Reviewer's Responses to Questions

**Comments to the Author**

1. If the authors have adequately addressed your comments raised in a previous round of review and you feel that this manuscript is now acceptable for publication, you may indicate that here to bypass the “Comments to the Author” section, enter your conflict of interest statement in the “Confidential to Editor” section, and submit your "Accept" recommendation.

Reviewer #1: All comments have been addressed

2. Is the manuscript technically sound, and do the data support the conclusions?

Reviewer #1: Yes

3. Has the statistical analysis been performed appropriately and rigorously? 

Reviewer #1: I Don't Know

4. Have the authors made all data underlying the findings in their manuscript fully available?

Reviewer #1: Yes

5. Is the manuscript presented in an intelligible fashion and written in standard English?

Reviewer #1: Yes

6. Review Comments to the Author

Reviewer #1: This MS has improved considerably compared to its first draft. I have just a very few simple corrections to point, which follow below. I think that after those small amendments are done, the manuscript is apt to be published.

P. 3, l. 61: Specify here that Henochilus is endemic from eastern Brazil. Brazil is a big country and in addition, the area of occurrence of Chilobrycon was specified (Pacifc slope of northern Peru and Ecuador), so the logical step is to do the same with Henochilus.

P. 4, l. 89-91: This sentence is more appropriate for the acknowledgements, not here.

P. 6, l. 134: Triportheus, not Tryportheus.

P. 14, l. 335 and 337: Chilobrycon, not Chylobrycon.

7. PLOS authors have the option to publish the peer review history of their article (what does this mean?). If published, this will include your full peer review and any attached files.

Reviewer #1: No

---

## [Author Response · Author response to Decision Letter 1]

6 Aug 2024

Response to reviewers

Dear Editor,

We really appreciate the detailed revision of the reviewers and have edited the manuscript following your valuable recommendations, which have led to improve our paper.

We hope that the manuscript is now suitable for publication in Plos One.

PONE-D-24-08613R1

Exploring the mitochondrial genomes and phylogenetic relationships of trans-Andean Bryconidae species (Actinopterygii: Ostariophysi: Characiformes)

PLOS ONE

Dear Dr. Márquez,

Thank you for submitting your manuscript to PLOS ONE. After careful consideration, we feel that it has merit but does not fully meet PLOS ONE’s publication criteria as it currently stands. Therefore, we invite you to submit a revised version of the manuscript that addresses the points raised during the review process.

We look forward to receiving your revised manuscript.

Kind regards,

Roberto E. Reis, Ph.D.

Academic Editor

PLOS ONE

Journal Requirements:

Now, we have replaced the studies previously cited as references 1 and 2 with the following new sources:

1. Birky CW, Jr. Transmission genetics of mitochondria and chloroplasts. Annu Rev Genet. 1978; 12(1):471–512. doi:10.1146/ annurev.ge.12.120178.002351.

2. Harrison RG. Animal mitochondrial DNA as a genetic marker in population and evolutionary biology. Trends Ecol Evol. 1989; 4(1): 6–11. doi:10.1016/0169-5347(89)90006-2.

Additional Editor Comments:

This revised version of the ms in considerably improved compared to its first draft. Please follow the corrections by Reviewer#1 below and resubmit for acceptance.

Reviewers' comments:

Reviewer's Responses to Questions

Comments to the Author

1. If the authors have adequately addressed your comments raised in a previous round of review and you feel that this manuscript is now acceptable for publication, you may indicate that here to bypass the “Comments to the Author” section, enter your conflict of interest statement in the “Confidential to Editor” section, and submit your "Accept" recommendation.

Reviewer #1: All comments have been addressed

2. Is the manuscript technically sound, and do the data support the conclusions?

Reviewer #1: Yes

3. Has the statistical analysis been performed appropriately and rigorously?

Reviewer #1: I Don't Know

4. Have the authors made all data underlying the findings in their manuscript fully available?

Reviewer #1: Yes

5. Is the manuscript presented in an intelligible fashion and written in standard English?

Reviewer #1: Yes

6. Review Comments to the Author

Reviewer #1: This MS has improved considerably compared to its first draft. I have just a very few simple corrections to point, which follow below. I think that after those small amendments are done, the manuscript is apt to be published.

P. 3, l. 61: Specify here that Henochilus is endemic from eastern Brazil. Brazil is a big country and in addition, the area of occurrence of Chilobrycon was specified (Pacifc slope of northern Peru and Ecuador), so the logical step is to do the same with Henochilus.

Done

P. 4, l. 89-91: This sentence is more appropriate for the acknowledgements, not here.

Done

P. 6, l. 134: Triportheus, not Tryportheus.

Done

P. 14, l. 335 and 337: Chilobrycon, not Chylobrycon.

Done

7. PLOS authors have the option to publish the peer review history of their article (what does this mean?). If published, this will include your full peer review and any attached files.

Do you want your identity to be public for this peer review? For information about this choice, including consent withdrawal, please see our Privacy Policy.

Reviewer #1: No

Done

---

## [Editor Report · Decision Letter 2]

8 Aug 2024

Exploring the mitochondrial genomes and phylogenetic relationships of trans-Andean Bryconidae species (Actinopterygii: Ostariophysi: Characiformes)

PONE-D-24-08613R2

Dear Dr. Márquez,

We’re pleased to inform you that your manuscript has been judged scientifically suitable for publication and will be formally accepted for publication once it meets all outstanding technical requirements.

Kind regards,

Roberto E. Reis, Ph.D.

Academic Editor

PLOS ONE

---

## [Editor Report · Acceptance letter]

15 Aug 2024

PONE-D-24-08613R2 

PLOS ONE

Dear Dr. Márquez, 

I'm pleased to inform you that your manuscript has been deemed suitable for publication in PLOS ONE. Congratulations! Your manuscript is now being handed over to our production team.

Kind regards, 

on behalf of

Dr. Roberto E. Reis 

Academic Editor

PLOS ONE